# Dermatophytomas: Clinical Overview and Treatment

**DOI:** 10.3390/jof8070742

**Published:** 2022-07-19

**Authors:** Aditya K. Gupta, Tong Wang, Elizabeth A. Cooper

**Affiliations:** 1Division of Dermatology, Department of Medicine, University of Toronto, Toronto, ON M5S 3H2, Canada; 2Mediprobe Research Inc., 645 Windermere Road, London, ON N5X 2P1, Canada; twang@mediproberesearch.com (T.W.); lcooper@mediproberesearch.com (E.A.C.)

**Keywords:** onychomycosis, antifungals, dermatophytoma

## Abstract

Dermatophytomas are characterized as a hyperkeratotic fungal mass in the subungual space, showing as dense white or yellow, typically in longitudinal streaks or patches. Masses can be visualized by traditional microscopy or histology. Newer technologies such as dermoscopy and optical coherence tomography also provide visual features for dermatophytoma diagnosis. The density of fungal mass, and lack of adherence to the nail structures, as well as possible biofilm development, may play a role in the reduction in drug penetration and subsequent lack of efficacy with traditional oral therapies such as terbinafine and itraconazole. A combination of drug treatment with mechanical or chemical debridement/avulsion has been recommended to increase efficacy. The topical antifungal solutions such as tavaborole, efinaconazole, and luliconazole may reach the dermatophytoma by both the transungual and subungual routes, due to low affinity for keratin and low surface tension. Current data indicates these topicals may provide efficacy for dermatophytoma treatment without debridement/avulsion. Similarly, fosravuconazole (F-RVCZ) has an improved pharmacological profile versus ravuconazole and may be an improved treatment option versus traditional oral therapies. The availability of improved treatments for dermatophytomas is crucial, as resistance to traditional therapies is on the increase.

## 1. Introduction

Dermatophytomas were first described in 1998 by Roberts and Evans [1]. This form of onychomycosis has been considered rare but may be underdiagnosed due to co-presentation with distal and lateral subungual onychomycosis (DLSO), total dystrophic onychomycosis (TDO), white superficial onychomycosis (deep form) or other types of onychomycosis [2]. A dermatophytoma manifests as a white or yellow round mass, spike, or streak that represents a hyperkeratotic aggregation of fungal filaments and spores, with the most active areas possibly adhering to the ventral surface of the nail plate, but with little adherence to the nail bed [1,3,4].

The name “dermatophytoma” is derived from “dermatophyte”, a class of fungus that infects human skin with “derma-” meaning skin and “-phyte” meaning plants, and “-oma” meaning masses or tumors. A dermatophytoma is similar to an aspergilloma; the latter is a densely packed mass of fungal filaments or “ball” of aspergillus organisms that may be found in the lung cavity, and possibly other body cavities [5,6,7]. Dermatophytomas have shown poor antifungal cure rates and thus are excluded from most clinical trials [8,9,10].

## 2. Clinical Features

The clinical presentation can be somewhat variable, and the distinction between dermatophytoma and distal and lateral subungual onychomycosis, extensive white superficial (deep variety), and extensive proximal subungual onychomycosis may not always be clear-cut [11,12]. Dermatophytomas typically present as opaque yellow, orange, or white patches in the nail plate or subungual region. They may also manifest as longitudinal streaks or spikes found along the lateral edges of the nail plate that extend from the distal edge or differently sized bands and patches (Figure 1) [1,3,4]. The initial development of this condition may occur at or near the hyponychium, typically seen with streaks along the lateral nail folds extending proximally to the lunula, or in patches that do not connect with the distal free edge of the nail [4,9].

Due to a lack of awareness, dermatophytoma may be misdiagnosed as onycholysis, potentially causing a delay in patients receiving the appropriate antifungal treatment. This may result in the exacerbation of onychomycosis and possible nail deformity [4,13].

Areas of the nail unit with underlying dermatophytomas are onycholytic, caused by the growing hyperkeratotic mass composed of fungal elements and debris [1,14]. Although the precise risk factors are unclear, worsened clinical prognosis due to dermatophytoma has been reported, likely due to onycholysis and poor adherence to the nail bed leading to reduced drug penetration into the fungal mass.

### Altered Metabolism and Possible Biofilms

Densely packed fungal filaments and large spores including resting chlamydospores and arthroconidia in the subungual space may provide altered metabolic activity in dermatophytomas versus other fungal presentations and may show abnormal presentation in microscopy/histology exam [1]. Increased dormancy and fungal density may provide resistance to antifungal penetration.

Recent investigations have shown that such microbial “masses” may represent a “biofilm” structure. Biofilms consist of microbial populations or communities massed together and surrounded by an extracellular matrix that forms a protective layer [15]. Biofilms could explain the low oral treatment efficacy of dermatophytomas, as a biofilm may protect the mass from physical disruptions, reduce drug penetration, alter metabolic activities and gene expressions of the mass, as well as allow evasion from host immune detection. *Trichophyton rubrum*, other dermatophytes, and non-dermatophyte molds (NDMs) have been shown to form biofilms in vitro [16,17,18,19,20].

## 3. Diagnosis

There are no standardized protocols for diagnosing dermatophytoma. Traditional diagnosis has mainly relied on clinical observations of a fungal “mass”. However, potassium hydroxide (KOH) microscopy and fungal culture remain essential to demonstrating dermatophyte presence, with or without histological examinations [1,2,21,22,23]. Increased detection of dermatophytoma may be reported in the future with the use of improved visualization technology such as dermoscopy and optical coherence tomography.

### 3.1. Mycology/Histology

*Trichophyton rubrum* has been identified as the most common causative organism in dermatophytoma patients [22,24]. Other causative organisms include *T. mentagrophytes*, *Epidermophyton floccosum*, non-dermatophyte molds (NDMs), and *Candida* species [21,24]. A direct microscopic exam may show the presence of a dense adherent mass of hyphae and spores under the nail plate. Production of arthroconidia may mediate the growth of these fungal masses, especially in the case of *T. rubrum*. Histological examination of the affected nail plate, typically performed using the periodic acid–Schiff stain, has shown either branching dermatophyte hyphae or fungal elements in a densely compacted mass [21,22]. The hyphae may be thick-walled and somewhat abnormal appearing [1].

### 3.2. Dermoscopy

Dermoscopy is a hand-held device that utilizes epiluminescence microscopy for close examination of subungual structures. As opposed to other diagnostic methods such as fungal cultures or histological staining, dermoscopic analysis is a point-of-care method; it has less cost and could fast-track treatment initiation in appropriate instances [25]. This method can differentiate patients with and without onychomycosis, including those complicated by dermatophytomas confirmed by laboratory tests [25,26,27,28]. Distinct microscopic features were identified, including the aurora borealis pattern, longitudinal streaks, and spikes in onychomycosis patients, as well as discolored subungual regions with longitudinal bands extending proximally in dermatophytomas [29,30].

### 3.3. Optical Coherence Tomography (OCT)

Based on the principle of interferometry, OCT is a novel application that utilizes low-coherence light for cross-sectional imaging of nails in vivo [31,32]. This technique was first utilized by *Verne* et al. to differentiate nails of onychomycosis patients with and without dermatophytomas, as well as nails of healthy individuals [33]. Nails with dermatophytoma exhibited a hyperreflective jagged border with a distinct and solid demarcation underneath the ventral nail plate and above the nail bed, which represents the characteristic fungal aggregates seen in these patients. The avascular fungal mass could be differentiated from the underlying vascular nail bed. These OCT findings were distinct from healthy nail presentation and distal subungual onychomycosis. The non-invasive nature of this method, combined with a shortened duration as opposed to fungal cultures, offers an easy, quick, and sensitive alternative to the traditional diagnostic methods.

## 4. Current Treatment Options

Due to the difficulty of drug penetration through the compacted fungal mass, the traditional oral agents alone have not typically been curative for dermatophytomas. Instead, surgical removal of the nail plate, either through targeted debridement or avulsion, combined with topical or oral treatment has been suggested to improve efficacy [1,13,15,22,34,35]. Removing the nail plate entirely through chemical debridement (for example, 40% urea cream under occlusion) or mechanical debridement/avulsion could be another option [21]. Martinez-Herrera et al. described seven cases of dermatophytomas but did not have any outcomes to report following chemical avulsion using 40% urea and 1% bifonazole [2].

There is scarce literature on the efficacy of nail removal alone compared to combination treatments with topical or oral antifungals. Such treatment will expose the compacted fungal elements and possible biofilms in the subungual space, which could then be removed using a curette [36]. The hyperkeratotic mass is not particularly adherent to the nail bed or overlying the ventral aspect of the nail plate; it can generally be readily scraped off [1]. Further investigation of debridement and avulsion is needed to better understand the possible role of the physical reduction in the fungal burden on the efficacy of treatment.

As reports of terbinafine resistance increased, the need for alternative therapies also increased. In contrast to the traditional therapies, the latest generation of topical therapies, (e.g., tavaborole, efinaconazole, and luliconazole) designed with improved transungual nail plate penetration chemistry as well as direct subungual penetration have shown promising efficacy for dermatophytomas without the need for concomitant debridement or avulsion (Table 1) [24,28,37,38,39]. Fosravuconazole is currently the only product of the newer generation of oral antifungals tested for dermatophytomas. Data on these treatments tend to come from retrospective analyses; more investigation for dermatophytomas is needed for these next-generation products (Table 1).

### 4.1. Topical Treatment—Case Reports

Several case reports indicate that topical therapy with efinaconazole 10% solution has been effective for dermatophytoma treatment. Cantrell et al. reported the first case of a dermatophytoma responding to efinaconazole 10% solution with resolution within 3 months of efinaconazole application [39]. Noguchi et al. reported on a dermatophytoma that had terbinafine-resistant *Trichophyton interdigitale* [28]. The organism was determined to be sensitive to efinaconazole in vitro, with minimum inhibitory concentration (MIC) to terbinafine of 0.5 µg/mL and efinaconazole MIC ≤ 0.015 µg/mL. Application of efinaconazole 10% solution resulted in complete cure after 10 months. No nail debridement was required.

Ciclopirox nail lacquer 8% solution, initially approved for treating onychomycosis in the USA in 1999, has shown efficacy in treating mild to moderate cases of DLSO. Ciclopirox has a multifaceted mode of action that disrupts fungal metabolism, including the chelation of trivalent cations that inhibit the key enzymes during mitochondrial electron transport, as well as inhibition of peroxidases and catalases [36,43]. Recently, a case report was published detailing the use of topical ciclopirox olamine 1% solution that successfully resolved a dermatophytoma after five months, when used in combination with mechanical nail debridement [44]. The potential efficacy of ciclopirox in treating dermatophytomas warrants further investigation.

Larger-scale investigations of tavaborole, efinaconazole, and luliconazole have since been reported, with good rates of dermatophytoma resolution (Table 1).

### 4.2. Tavaborole Topical Solution

Tavaborole 5% solution, initially approved in the USA in 2014 for mild to moderate dermatophyte onychomycosis, disrupts fungal protein synthesis through the inhibition of leucyl tRNA synthase.

In a post hoc evaluation of topical tavaborole in North America (USA and Mexico) 102 cases of dermatophytoma were identified out of the 360 enrolled subjects [40]. Tavaborole doses from 1% to 7.5%, in a variety of frequencies, were tested against vehicles in three Phase II studies. For the approved dosage of tavaborole 5%, clearance of dermatophytomas at Day 360 was found in 28.2% (11/39) of subjects compared to 6.3% (1/16) subjects applying vehicle (Table 1). Overall, for any tavaborole dosing regimen, complete resolution of the dermatophytoma was reported in 26.7% (23/86) subjects (Table 1).

Resolution of dermatophytomas at Day 180 of a tavaborole regimen was found in 24.4% (21/86) patients compared to 0% of the vehicle-treated patients; 13 of 19 (68.4%) subjects from the 21 cured patients remained clear at Day 360. In vitro data demonstrated that tavaborole is retained in the nail at levels exceeding the minimum inhibitory concentration of *T. rubrum* and *T. mentagrophytes* for 90 days post-dosing (following once daily application for 28 days) [40,45].

### 4.3. Efinaconazole 10% Solution

Despite issues of drug penetration with topical treatments, efinaconazole 10% solution, a triazole antifungal targeting ergosterol biosynthesis causing cell membrane disruption, has shown promising results albeit with a variable success rate (<25% to 100%) with once-daily treatment duration up 48 weeks [24,38,39]. The efficacy of the efinaconazole solution may be related to the high degree of penetration through the transungual route. There is also evidence of the spread of drugs through the subungual route [46,47,48]. This results in a high fungicidal concentration of the drug at the site of the dermatophytoma. Penetration is aided by the low keratin affinity and low surface tension of the efinaconazole solution.

Wang et al. report on an open study where efinaconazole 10% solution was used to treat distal and lateral onychomycosis (DLSO) complicated with a dermatophytoma (19 patients with 20 target nails) [24]. The diagnosis was made clinically, and a majority of patients also demonstrated positive mycology (KOH positive or culture positive for a dermatophyte). All patients applied efinaconazole 10% solution once daily for at least 48 weeks. All subjects had resolution of dermatophytomas, though only 63% (12/19 patients) also had a clearance of DLSO and dermatophytomas (Table 1). The mean time to resolution of the dermatophytomas was 16 weeks (range 4–24 weeks). All cultures were negative at the end of treatment (week 52) and end of study (week 56).

In the series by Shimoyama et al., efinaconazole 10% was effective in treating three of five dermatophytomas after a mean treatment duration of 14.8 months (Table 1) [37].

Watanabe et al. conducted a post hoc retrospective study to evaluate the efficacy of once-daily efinaconazole 10% solution for up to 72 weeks to treat onychomycosis with longitudinal spikes [38]. The causative fungal species were *T. rubrum* 51/82 (62.2%), *T. interdigitale* 19/82 (23.2%), and other *Trichophyton* species 12/82 (14.6%). At the end of 72 weeks of treatment, the disappearance of spikes was noted in 81.7% (67/82) of treated patients (Table 1). The complete cure of all onychomycoses (0% DLSO/dermatophytomas and negative KOH result) was 41.5% (34/82) (Table 1). The mycologic cure rate was 59/82 (71.9%). The authors postulate that the efficacy of the efinaconazole solution is related to the high degree of penetration through the transungual route.

### 4.4. Luliconazole 5% Nail Solution

There is limited data on the efficacy of luliconazole in treating dermatophytomas. It is an imidazole antifungal that disrupts fungal cell membranes via inhibition of ergosterol biosynthesis, similar to efinaconazole. The 5% solution has been approved in Japan for the treatment of onychomycosis.

A retrospective review of patients treated with topicals showed that luliconazole 5% solution was effective in treating three of six dermatophytomas after a mean treatment duration of 11.2 months [37].

### 4.5. Fosravuconazole

Fosravuconazole L-lysine ethanolate (F-RVCZ is a pro-drug that is metabolized into the triazole antifungal ravuconazole. F-RVCZ has significantly increased bioavailability and efficacy compared to the ravuconazole structure, with fewer drug interactions than standard oral treatments [42]. F-RVCZ is currently only approved for onychomycosis in Japan, but available data indicates it may be a suitable alternative for dermatophytomas where oral therapy is desired.

F-RVCZ has shown efficacy in treating dermatophytomas. In a case report by Suzuki et al., a patient with concurrent tinea faciei, tinea corporis, and tinea unguium complicated by dermatophytomas was given a daily regimen of F-RVCZ at 100mg for 3 months. After one year, there was a notable improvement of deformities on the great toenails, without significant side effects on hepatic function [23].

In the study by Shimoyama, 109 patients with onychomycosis of varying types were treated with fosravuconazole one capsule per day (equivalent to 100 mg of ravuconazole for 12 weeks (Table 1) [42]. Dermatophytomas were present in 21 patients based on a retrospective photo review of the treated subjects. Complete clearance with negative microscopy findings was found in 12/21 (57.1%) patients. The mean improvement rate of the affected nail area at the last visit was 79.8 ± 28.9%. The average observation period to complete cure was 34.3 ± 11.1 weeks.

A good safety profile was noted in the overall treated patient population. Of the 109 patients, there were 6 patients (5.5%) patients who discontinued therapy. Three patients (2.8%) discontinued therapy due to elevated liver function tests (LFTs) (mean serum gamma-glutamyl transferase 547.7 ± 679.3 IU/L, aspartate aminotransferase 70 ± 15.1 IU/L, and alanine aminotransferase 129.3 ± 59.2 IU/L). The other three patients discontinued therapy due to abdominal discomfort. The adverse events improved following the discontinuation of therapy. The mean administration period in the patients who discontinued therapy was 6.3 ± 2.0 weeks [42].

## 5. Summary

Dermatophytomas were first described about 25 years ago by Roberts and Evans [1]. Though traditionally considered a “spike”, the appearance of dermatophytomas may vary considerably. They are characterized as a hyperkeratotic fungal mass in the subungual space, showing as dense white or yellow in longitudinal streaks or patches. The clinical suspicions can be confirmed by traditional microscopy or histology, with newer technologies such as dermoscopy and optical coherence tomography also able to provide visual features for dermatophytoma diagnosis.

The density of fungal mass, and lack of adherence to the nail structures, as well as possible biofilm development, may play a role in the reduction in drug penetration and subsequent lack of efficacy with traditional oral therapies such as terbinafine and itraconazole. A combination of drug treatment with mechanical or chemical debridement/avulsion has been recommended to increase efficacy.

The topical antifungal solutions such as efinaconazole 10% solution or tavaborole 5% solution may reach the dermatophytoma by both the transungual and subungual routes. They may achieve high fungicidal concentrations in the dermatophytoma; this is facilitated by their low affinity for keratin and low surface tension. Similarly, F-RVCZ has an improved pharmacological profile versus ravuconazole and may be an improved oral treatment option versus traditional therapies. The availability of improved treatments for dermatophytomas is crucial, as resistance to traditional therapies is on the increase.

## Figures and Tables

**Figure 1 jof-08-00742-f001:**
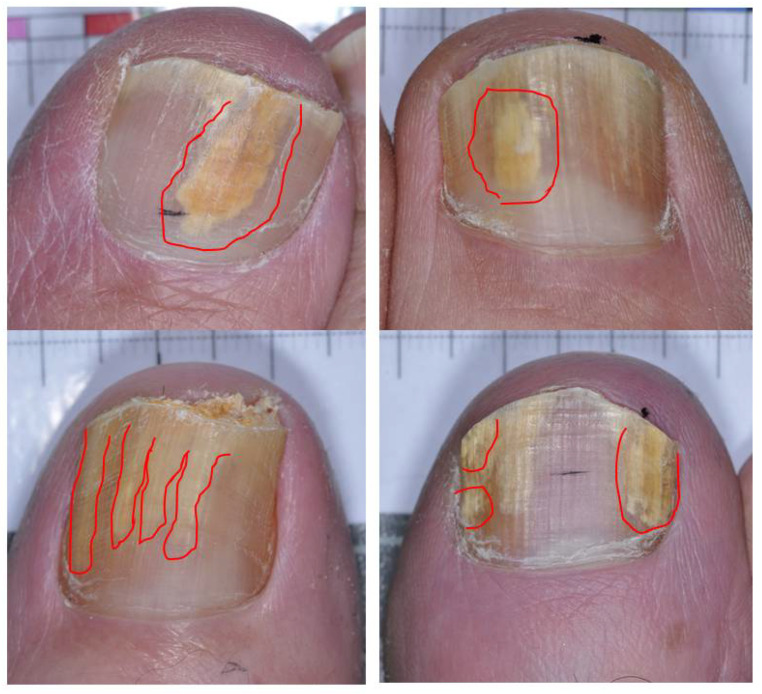
Dermatophytoma presentations in great toenails with culture-confirmed dermatophytosis.

**Table 1 jof-08-00742-t001:** Summary of the newer therapies for dermatophytoma treatment.

Source	Study Design	Treatment Regimen(s) Treatment	Outcome (Definition)	Outcome Rate, % (n/Total)
**Topical Tavaborole** topical solution (“TAV”) Aly et al., 2018 [40]	3 phase II studies: post hoc photo review N = 102 (TAV = 86 pts; vehicle = 16 pts) - 18–65 years of age - Onychomycosis in at least one great TN - 20–60% nail area at baseline	(a) Vehicle OD for 90d then 3× wkly for 90 d (Study# 200)	Resolution = no clinical features of dermatophytoma (well-defined yellow or white patches or streaks) at Day 360 (“1 year”)	(a) 6.3% (1/16)
		(b) TAV 1% OD for 180 d (Study# 203)		(b) 33.3% (1/3)
		(c) TAV 2.5% OD for 90 d then 3× wkly for 90 d (Study# 200)		(c) 18.2% (2/11)
		(d) TAV 5% OD for 90 d then 3× weekly for 90 d (Study# 200)		(d) 30.0% (3/10)
		(e) TAV 5% OD for 180 d (Study# 201)		(e) 27.3% (3/11)
		(f) TAV 5% OD for 360 d (Study# 201)		(f) 20.0% (2/10)
		(g) TAV 5% OD for 30 d, then 3× weekly for 150 d; FU at 360 d (Study# 203)		(g) 37.5% (3/8)
		(h) TAV 7.5% OD for 90 d then 3x weekly for 90 d (Study# 200)		(h) 20.0% (5/25)
		(i) TAV 7.5% OD for 180 d (Study# 201)		(i) 50.0% (4/8)
		**Any TAV regimen**		26.7% (23/86)
		**Any 5% TAV regimen**		28.2% (11/39)
**Topical Efinaconazole** 10% solution Wang et al., 2019. [24]	N = 19 pts (20 TN dermatophytomas) - ≥18 years of age - Presence of DLSO with dermatophytoma, on one or both great toenails - Nail thickness ≤ 3 mm - Positive KOH test or culture positive for dermatophyte (12/19 pts culture positive at baseline)	EFN once daily for at least 48 weeks	Clinical resolution of dermatophytoma at end of study (week 56 or ET)	100% (20/20)
			“Cure”: ≤10% affected area DLSO/dermatophytoma and negative culture) at end of study (week 56 or ET)	63% (12/19 pts) 65% (13/20 TNs)
**Topical Efinaconazole** 10% solution Watanabe et al., 2021 [38]	Retrospective photo review N = 82 pts - Adult patients with great toenail dermatophyte onychomycosis, showing longitudinal spikes based on images. - Original patient population from Iozumi et al. [41]	Apply over affected nail once daily for up to 72 weeks	“Disappearance of spikes” at wk 72 or last visit	81.7% (67/82)
			“Complete Cure” at wk 72 or last visit = 0% involvement (DLSO and spikes cleared) plus KOH negative	41.5% (34/82)
**Topical Efinaconazole** 10% solution Shimoyama et al., 2019. [37]	Retrospective chart/photo review 62 total patients: Efinaconazole 10% (N = 5, with dermatophytoma) -Patients diagnosed with onychomycosis in any nail by direct microscopic examination	Topical EFN for at least 1 month (N = 5 pts) (Mean treatment duration: 14.8 m)	“Complete Cure” at last visit = not defined; (clinical clearance only, no mycology discussed)	60% (3/5)
**Topical Luliconazole** 5% nail solution Shimoyama et al., 2019. [37]	Retrospective chart/photo review 72 total patients: Luliconazole 5% (N = 6, with dermatophytoma) - Patients diagnosed with onychomycosis in any nail by direct microscopic examination	Topical Luliconazole for at least 1 month (N = 6) (Mean treatment duration: 11.2 m)	“Complete Cure” at last visit = not defined; (clinical clearance only, no mycology discussed)	50% (3/6)
**Oral Fosravuconazole** Shimoyama et al., 2021. [42]	Retrospective photo review -Adult patients with onychomycosis in any nail treated with fosravuconazole between 1 March 2018–31 August 2020 confirmed by direct microscopy (N = 109) “Dermatophytoma” = microscopy showing abundant fungal filaments, large spores, or both, which were compacted and formed a mass or fungal ball (N = 21 of 109, 19.3%)	1 capsule (100 mg RAV equivalent)/day for 12 weeks	“Complete Cure” at wk 12 = 100% clearance and negative microscopy	0% (0/21)
			“Complete Cure” at last visit = 100% clearance and negative microscopy	57.1% (12/21)

Legend: TAV = Tavaborole; EFN = Efinaconazole; RAV = Ravuconazole; m = month; d = day; wk = week; wkly = weekly; OD = once daily; pts = patients; TN = toenail; ET = early termination; FU = follow-up.

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
