# Peer review of "Dermatophytomas: Clinical Overview and Treatment"

_jof, 2022, doi:10.3390/jof8070742_

Round 1

Reviewer 1 Report

Nice review with good discussion of antifungals. 

Author Response

Thank you for taking time for the review, and for your comments.

Reviewer 2 Report

The article present an overview of recent methods for curation of infected nails. The authors present interesting results for topical and oral therapies with new antifmycotica for the treatment of onychomycosis.

The article is written clearly and give important hints for treatment parameters.

Minor problem:

Title and whole text: the term “dermatophytoma”

In Introduction, (Lines33-35) the different parts of the word were explained nicely.  Nevertheless, the authors mediate the impression that fungi were linked with plants. Fungi represent their own kingdom and represents the sister group of Metazoa (multi-cellular animals). 

In my opinion, it is better to use a fungal terminology and change dermatophytomas into dermatomycetomas and dermatophytes into dermatomycetes. The use of plant terminology has only historical reasons and is not supported by molecular phylogeny.

Physiological pathways like the sterol biosynthesis show a high degree of similarity between fungi and animals. Therefore, the development of antifungals is difficult and be an explanation for side effects of this substances to the host.

Author Response

We agree it is unfortunate that the roots of the term ‘dermatophyte’ and ‘dermatophytoma’ imply plants, but in clinical medicine these are the most-used terms, specifically implying fungi associated with skin and nails, and the terms are well-understood as fungi rather than plants for clinicians. Use of a ‘mycetoma’ root name will unfortunately cause confusion for clinicians with the condition ‘mycetoma’ which is a deep fungal infection of skin quite different from tinea infection of skin and nails. At this time there is no ideal term to adopt and the authors will continue to use the term ‘dermatophytoma’.

Reviewer 3 Report

The manuscript jof-1798038 summarizes the recent findings concerning the antifungal therapy of dermatophytomas.

The manuscript offers a balanced, comprehensive and critical view on the available literature data regarding the diagnosis, current treatment options and efficacy of antifungals in the treatment of dermatophytomas. The manuscript is generally well structured with a critical report of the findings of available clinical studies (including an extensive and rather interesting table summarizing these clinical studies).

The authors focus on topical antifungal solutions which provide higher efficacy for dermatophytomas treatment without nail debridement or avulsion. More, fosravuconazole has been prompted as an improved treatment option as compared to traditional oral therapies.

I consider this overview useful to clinicians and also to the scientific community interested in antifungal research and antifungal therapy, considering the high rates of microbial resistance to conventional antifungals.

Specific comment: ”Page 2, Line 73: Please use full name for T. rubrum, as it is the first mention in the text.”

Author Response

Thank you for taking time for the review, and for your comments. The full spelling of T. rubrum is now used in this line.

Reviewer 4 Report

Authors performed a comprehensive review on dermatophytomas which are a form of onychomycosis. Few is known on these entities, and is an interesting subject.

The manuscript is clear and well written. However I have some comments.

General comments: 

1) The manuscript structure has to be improved. It would be clearer for the reader to keep main sections such as "clinical features", diagnosis" and "treatment" and to include the subsections in these sections accordingly. 

2) The manuscript content is unbalanced, the part on antifungal treatment is much longer than the rest of the manuscript. The manuscript title should be modified accordingly by including a word on treatment (such as "an overview and current therapeutic options"

Specific comments:

3) Introduction: please specify if dermatophytomas are rare or common in case of onychomycosis

4) Figure 1: please put arrows to show clearly where are the dermatophytomas 

5) Diagnosis (lines 75-79): mycological examination and fungal culture are essential for the diagnosis of onychomycosis. Even if it is not sufficient to differentiate dermatophytomas from other forms of onychomycosis, direct examination can prove the ungual disease and culture is essential to obtain the identification of the species involved which can drive therapeutic options. Please modify your sentence accordingly.

6) Line 84: please change the expression "under the microscope" 

7) you say that dermatophytomas are like aspergillomas (line 35). In aspergillomas, fungal culture often fails as the fungus is mainly dead. Is it the same with dermatophytomas ?

8) Table 1 is confused and has to be improved: please detail acronyms such as FU, TNE or ET. Topical tavaborole last column, TAV percentages are confusing, please remove them. Please reorganize rows to not mix antifungal, especially efinaconazole and luliconazole.

9) Topical treatment - case reports: remove this subheading there is some points that are detailed later in the manuscript (such efinaconazole solution) 

Author Response

Thank you for taking time for the review, and for your comments. Our responses are below.

1 – manuscript structure

Authors: This manuscript in original format was set up as suggested by this reviewer – headers have been numbered and reformatted to more clearly designate the sections.

2 – title revision

Authors: The title has been modified to ‘Dermatophytomas: Clinical Overview and Treatment’ to capture the strong emphasis on treatment and also to address the more clinical nature of the manuscript discussion rather than primary fungal biology emphasis.

3 – dermatophytoma rare or common

Authors: Dermatophytomas are ‘rare’ to date, but likely underdiagnosed. This information has been added to the introduction. An additional sentence in ‘diagnosis’ section has now also mentions that new technology may increase detection of dermatophytomas.

4 – put arrows on dermatophytomas in photos

Authors: Dermatophytoma outlines are now added in red on the photos.

5 – diagnosis – modify re: microscopy/culture

Authors: Visual assessment is to note the ‘mass’ of dermatophytoma only; The authors agree that culture is essential to show a dermatophyte fungus infection and we have modified the sentences accordingly to better reflect these points.

6 – modify ‘under the microscope’

Authors: The sentence has now been updated.

7 – per aspergilloma, is dermatophytoma mostly ‘dead’?

Authors: This is likely correct for the dermatophytoma as well. We have renamed the ‘biofilm’ section to ‘Altered metabolism and Biofilms’; fungal dormancy is discussed here.

8 – Table 1 clarifications suggested

Authors: Per reviewer comments, legend now includes missing abbreviations, tavaborole percentages have been removed and rows are reorganized to separate the Efinaconazole and Luliconazole study data.

9 – re; case report/subheading

Authors: Case reports is a relevant subsection, as this type of data is generally not weighted as highly as formal clinical trials. Also the case report data has not been included in the tables, so we did not wish to include this data with the clinical trial data reporting sections that follow the table. Subsections are now reformatted to more clearly to reflect their position against the main headers.